# ProJo4D: Progressive Joint Optimization for Sparse-View Inverse Physics Estimation

**Daniel Rho**  *dnl03c1@cs.unc.edu*
*University of North Carolina at Chapel Hill*

**Jun Myeong Choi**  *chedgekr@cs.unc.edu*
*University of North Carolina at Chapel Hill*

**Biswadip Dey**  *biswa-dey@ieee.org*
*Meta Reality Labs*

**Roni Sengupta**  *ronisen@cs.unc.edu*
*University of North Carolina at Chapel Hill*

**Reviewed on OpenReview:** *https://openreview.net/forum?id=pqvVrqlXCZ*

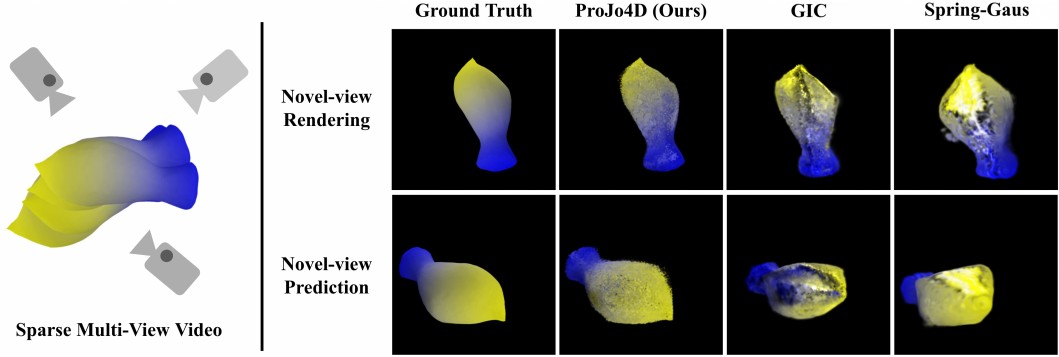

Figure 1: We present ProJo4D, a progressive joint optimization framework for estimating 4D representation and physical parameters of deformable objects from sparse multi-view video. ProJo4D significantly outperforms state-of-the-art inverse physics estimation algorithms, Spring-Gaus (Zhong et al., 2024) and GIC (Cai et al., 2024), which perform sequential optimization of scene geometry and physical parameters.

## Abstract

Neural rendering has advanced significantly in 3D reconstruction and novel view synthesis, and integrating physics into these frameworks opens new applications such as physically accurate digital twins for robotics and XR. However, the inverse problem of estimating physical parameters from visual observations remains challenging. Existing physics-aware neural rendering methods typically require dense multi-view videos, making them impractical for scalable, real-world deployment. Under sparse-view settings, the sequential optimization strategies employed by current approaches suffer from severe error accumulation: inaccuracies in initial 3D reconstruction propagate to subsequent stages, degrading physical state and material parameter estimates. On the other hand, simultaneous optimization of all parameters fails due to the highly non-convex and often non-differentiable nature of the problem. We propose ProJo4D, a progressive joint optimization framework that gradually expands the set of jointly optimized parameters. This design enables physics-informed gradients to refine geometry while avoiding the instability of direct joint optimization over all parameters. Evaluations on synthetic and real-world datasets demonstrate that ProJo4D substantially outperforms prior work in 4D future state prediction and physical parame-

ter estimation, achieving up to $10\times$ improvement in geometric accuracy while maintaining computational efficiency.

# 1 Introduction

Neural rendering techniques have made significant progress in 3D scene reconstruction and novel view synthesis (Mildenhall et al., 2020; Müller et al., 2022; Kerbl et al., 2023), but they often lack adherence to the underlying physical laws (e.g., conservation of energy, momentum, or monotonicity constraints). This gap severely restricts their usage in downstream applications that require not only photorealistic appearance but also physically plausible behavior (Chu et al., 2022). For instance, in vision-based robot learning (Li et al., 2024; Jiang et al., 2025; Abou-Chakra et al., 2024), agents trained in simulations must seamlessly transfer the learned skills to the real world, which requires accurate physical interactions within the synthetic environment. Similarly, XR applications in many engineering and industrial settings require rendered objects to respond meaningfully to user interactions (e.g., changes in material properties or object dimensions), environmental constraints, and external forces to maintain immersion, usability, and seamless integration of virtual and physical worlds (Jiang et al., 2024; Zheng et al., 2025).

A recent body of work (Cai et al., 2024; Li et al., 2023b; Zhong et al., 2024; Chen et al., 2025) has attempted to bridge this gap by incorporating physics-based priors into neural rendering pipelines. However, the state-of-the-art approaches typically rely on dense multi-view setups, often requiring more than ten synchronized cameras with known poses. Such instrumentation imposes significant practical barriers, particularly in scenarios demanding scalable, flexible, or in-situ data collection. Whether in robot learning or industrial XR applications, the ability to create physically plausible digital twins from sparse observations is critical for real-world deployment. Overcoming the dependency on dense multi-view capture is thus crucial to realizing the full potential of neural rendering in physically grounded, deployable systems.

Sparse-view settings pose significant challenges for accurate 3D reconstruction and physical property estimation due to occlusions, shape ambiguities, and limited viewpoints. Existing methods that excel under dense observations degrade markedly when faced with sparse inputs, primarily due to the accumulation of errors in their sequential optimization pipelines (Li et al., 2023b; Zhang et al., 2024; Huang et al., 2024; Zhong et al., 2024; Cai et al., 2024; Liu et al., 2025). These sequential optimization pipelines typically begin by learning an initial 3D or 4D scene representation from sparse images, which is often noisy and ambiguous, particularly in estimating geometry or particle positions. This flawed representation then serves as the basis for inferring initial physical states (e.g., initial velocities) and subsequently material properties (e.g., stiffness, Poisson's ratio). As a result, errors introduced early propagate and compound, ultimately degrading both physical state and material parameter estimates. Although some recent works (Zhong et al., 2024) have explored partial joint optimization of certain parameter subsets, they fall short of addressing the complete inverse physics problem from the outset. Fully joint optimization of all parameters remains challenging due to the highly non-convex, partly non-differentiable nature of the problem (Zhong et al., 2021), often leading to poor local minima, particularly under sparse views.

While prior work has focused on improving scene representations and physical models, the optimization strategy itself remains underexplored despite its critical role in sparse-view settings. Unlike sequential optimization strategies that optimize parameters one stage at a time, our progressive joint optimization gradually expands the set of jointly optimized parameters. This maintains coupling between geometry and physical parameters throughout optimization, preventing error accumulation while managing the non-convex optimization landscape. Progressive optimization succeeds where both alternatives fail. Unlike sequential methods, it allows physics-informed gradients to correct geometry errors throughout training. Unlike full joint optimization, it avoids the instability that arises from optimizing all parameters simultaneously on non-convex landscapes. As demonstrated in our experiments (Sec. 4), this progressive strategy achieves improved performance compared to both fully sequential and fully joint approaches across diverse materials and sparse-view settings.

To demonstrate the effectiveness of our progressive joint optimization strategy, without any model changes, we utilize GIC's 4D scene representation (Cai et al., 2024) and physical models, focusing only on changing

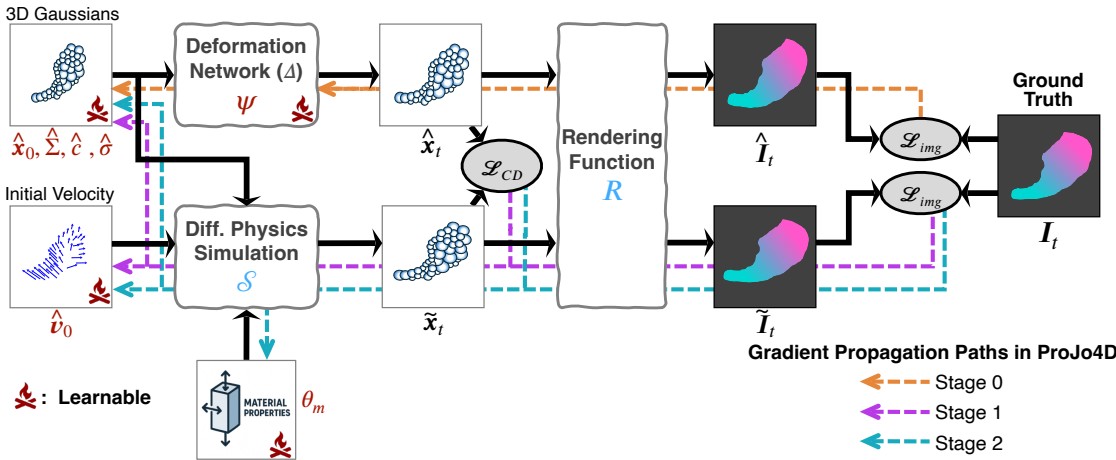

Figure 2: ProJo4D progressively grows the set of optimized variables (3D Gaussian parameters, deformation network, initial velocity, and material properties) across training stages to mitigate error propagation common in sequential frameworks like PAC-NeRF (Li et al., 2023b) and GIC (Cai et al., 2024). The diagram illustrates the inter-dependencies among parameters in the inverse physics estimation task, with colored dotted arrows indicating gradient flow during each optimization stage.

the optimization strategy. Through extensive evaluations, we show that our progressive joint optimization strategy significantly improves performance for inverse physics estimation on both synthetic and real-world datasets, mitigating the drastic performance drop in sparse-view scenarios. Our method outperforms the sequential baseline GIC (Cai et al., 2024), achieving strong results in 4D future state prediction (Chamfer Distance $16.11 \rightarrow 1.60$), future state rendering (PSNR $17.58 \rightarrow 22.30$), and physical parameter estimation (Poisson's ratio MAE $0.23 \rightarrow 0.10$, Young's modulus MAE $0.18 \rightarrow 0.09$), as shown in Table 2. While recent work MASIV (Zhao et al., 2025) achieves competitive results through neural constitutive models, this approach requires significantly longer optimization time. In contrast, ProJo4D uses explicit physical parameters and generally outperforms MASIV while maintaining lower computational requirements and physically interpretable parameters crucial for downstream applications.

To summarize, our contributions are as follows.

- We identify optimization strategy as a key bottleneck for sparse-view inverse physics estimation, showing that error accumulation in sequential pipelines severely degrades performance.
- We propose progressive joint optimization, a simple yet effective strategy that gradually expands the set of jointly optimized parameters, enabling physics-informed gradients to refine geometry while avoiding optimization instability.
- We demonstrate that this optimization strategy alone yields up to $10\times$ improvement in geometric accuracy across multiple datasets, without any architectural modifications.

## 2 Background

### 2.1 Notation and Problem Formulation

We introduce the key notations and define the inverse physics estimation problem we aim to solve.

**Input Data.** We are given a set of input images $I = \{I_{t,c}\}_{t \in \mathcal{T}, c \in \mathcal{C}}$, where $I_{t,c}$ denotes an image captured at time $t$ from camera $c$. For each image $I_{t,c}$, the corresponding camera pose $P_c$ from a predefined set of cameras $c \in \mathcal{C}$ and the timestamp $t \in \mathcal{T}$ are assumed to be known. In addition, a transparency alpha map $\alpha_{t,c}$ is often used for initial 3D/4D reconstruction, either rendered or estimated using segmentation or matting.

**3D / 4D Representation.** Our scene representation is based on 3D Gaussian Splatting (Kerbl et al., 2023). The 3D Gaussians are parameterized by their initial positions $\hat{\mathbf{x}}_0$, covariance matrices $\hat{\Sigma}$, color features $\hat{c}$,

and opacity $\hat{\sigma}$. For representing 4D dynamics, we need additional parameters $\psi$. The position of a Gaussian at time $t$ is denoted by $\hat{\mathbf{x}}_t$ and is related to its initial position $\hat{\mathbf{x}}_0$ via a displacement function $\Delta(\cdot)$, which models the motion of the Gaussians over time, parameterized by $\psi$ as:

$$\hat{\mathbf{x}}_t = \hat{\mathbf{x}}_0 + \Delta(\hat{\mathbf{x}}_0, \psi, t). \tag{1}$$

We denote the rendering function, a differentiable splatting algorithm (Kerbl et al., 2023), by $R(\cdot, \cdot, \cdot, \cdot; \cdot)$. This function takes the Gaussian parameters $(\hat{\mathbf{x}}_t, \hat{\Sigma}, \hat{c}, \hat{\sigma})$ at time $t$ and the camera pose $P_c$ as input, and outputs a rendered image $\hat{I}$:

$$\hat{I}_{t,c} = R(\hat{\mathbf{x}}_t, \hat{\Sigma}, \hat{c}, \hat{\sigma}; P_c). \tag{2}$$

Similarly, $R_\alpha(\cdot, \cdot, \cdot; \cdot)$ denotes the alpha map rendering function and $\hat{\alpha}$ denotes the rendered alpha map. For the detailed rendering process, please refer to 3D Gaussian Splatting (Kerbl et al., 2023).

**Physics Parameters.** Throughout the paper, we refer to both the initial physical state $s$, such as initial velocity $v_0$, and material parameters $\theta_m$ as physical parameters. Material parameters $\theta_m$ include Young's modulus $E$ and Poisson's ratio $\nu$ for elastic objects. We assume that the material model, e.g., elastic or plastic, is known a priori, consistent with all other prior works (Li et al., 2023b; Cai et al., 2024).

A physics simulation model, denoted by $\mathcal{S}(\cdot, \cdot, \cdot, \cdot)$, is used to predict the state of the system over time. Given the initial positions $\hat{\mathbf{x}}_0$, initial velocity $\hat{\mathbf{v}}_0$, material parameters $\hat{\theta}_m$, the simulation outputs the predicted positions $\tilde{\mathbf{x}}_t$ and the corresponding rendered image $\tilde{I}_{t,c}$ at time $t$ as:

$$\tilde{\mathbf{x}}_t = \mathcal{S}(\hat{\mathbf{x}}_0, \hat{\mathbf{v}}_0, \hat{\theta}_m, t), \quad \tilde{I}_{t,c} = R(\tilde{\mathbf{x}}_t, \hat{\Sigma}, \hat{c}, \hat{\sigma}; P_c). \tag{3}$$

**Problem Formulation.** In summary, our task is an inverse estimation problem: given input images $I$, their corresponding camera parameters $P$, we aim to estimate the underlying geometry $\mathbf{x}_0$, appearance parameters $(\Sigma, c, \sigma)$, and physical properties $(v_0, \theta_m)$ of a deformable object.

## 2.2 Related Works

**Differentiable Physics Simulation.** Differentiable physics simulation is widely used to optimize and estimate physics-related parameters (Xu et al., 2019; Sanchez-Gonzalez et al., 2020; Hu et al., 2020; Geilinger et al., 2020; Zhong et al., 2021; Murthy et al., 2021; Wang et al., 2024). This forms the foundation of this research area, raising important considerations about which differentiable simulation frameworks to employ and how to design and schedule the optimization process. Among the commonly used simulation methods are the spring-mass model (Zhong et al., 2024) and the Material Point Method (MPM) (Jiang et al., 2016). MPM is a particle-based method capable of handling diverse deformable materials, such as elastic, plastic, and granular, through different material models. Differentiable MPM (Hu et al., 2020) makes the simulation pipeline differentiable via automatic differentiation, enabling gradient-based optimization of initial positions, velocities, and material parameters. While MPM itself can simulate multiple materials, existing inverse physics methods (Li et al., 2023b; Cai et al., 2024) typically assume a single material model with global parameters per object. Following the same problem setting, we use differentiable MPM as the physics simulator $\mathcal{S}(\cdot, \cdot, \cdot, \cdot)$. Simplicits (Modi et al., 2024) can be used for accelerated inverse physics (Chen et al., 2025), but only supports (hyper)elastic materials with simple gravity-only scenarios. Our progressive joint optimization shares conceptual similarities with curriculum learning (Bengio et al., 2009) and coarse-to-fine optimization, where easier subproblems are solved before harder ones. However, unlike typical curricula over data or model complexity, we design a curriculum over the parameter space itself.

**Physics-based Neural Rendering.** Recent neural rendering methods have increasingly incorporated physical priors to enable accurate estimation of scene dynamics and material properties directly from video observations (Li et al., 2023a; Yu et al., 2023; Kaneko, 2024; Xue et al., 2023; Qiao et al., 2022; Ma et al., 2021; Guan et al., 2022; Gao et al., 2025). These approaches aim to recover physically meaningful parameters, such as material properties, forces, and initial states, by coupling differentiable rendering with physics-based simulation.

Table 1: Optimization strategies of existing methods. X, A, S, and M denote positions, appearances, physical states, and material parameters. 0 denotes initial 3D/4D representation learning before physical parameter estimation. $\triangle$ denotes optional optimization, depending on the scene.

| Method | Param. | 0 | Stage 1 | 2 | 3 | 4 |
|---|---|---|---|---|---|---|
| PAC-NeRF (Li et al., 2023b) PhysDreamer (Zhang et al., 2024) | X | ✓ | | | | |
| | A | ✓ | | | | |
| | S | | ✓ | | | |
| | M | | | ✓ | | |
| Spring-Gaus (Zhong et al., 2024) | X | ✓ | | | | $\triangle$ |
| | A | ✓ | $\triangle$ | | $\triangle$ | $\triangle$ |
| | S | | | ✓ | | $\triangle$ |
| | M | | | | | ✓ |
| GIC (Cai et al., 2024) MASIV (Zhao et al., 2025) | X | ✓ | | | | |
| | A | ✓ | | | ✓ | |
| | S | | ✓ | | | |
| | M | | | ✓ | | |
| Vid2Sim (Chen et al., 2025) | X | ✓ | ✓ | | | |
| | A | ✓ | ✓ | | | |
| | S | | | | | |
| | M | ✓ | ✓ | | | |
| Ours | X | ✓ | ✓ | ✓ | | |
| | A | ✓ | ✓ | ✓ | | |
| | S | | ✓ | ✓ | | |
| | M | | | ✓ | | |

Among early attempts, PAC-NeRF (Li et al., 2023b) proposed a general framework that sequentially optimizes geometry and appearance, initial physical states, and material parameters. Spring-Gaus (Zhong et al., 2024) introduced a spring-mass formulation within the 3D Gaussian Splatting framework (Kerbl et al., 2023) to model dynamic deformations. Gaussian Informed Continuum (GIC) (Cai et al., 2024) further improves physical parameter estimation and future-state prediction by leveraging learned 4D representations to guide physically motivated 3D losses. Vid2Sim (Chen et al., 2025) incorporates pretrained models to initialize material parameters and adopts Simplicits (Modi et al., 2024) for accelerated per-scene optimization. MASIV (Zhao et al., 2025) extends this line by introducing neural material models to replace explicit constitutive laws, achieving material-agnostic representations through learned neural networks (Ma et al., 2023). While this architectural innovation enables flexibility across material types, it introduces additional model complexity and significantly increases optimization time. In contrast, our approach maintains the same scene representation and physics framework as prior work (Cai et al., 2024), achieving better performance purely through progressive joint optimization without architectural modifications.

Our method focuses on accurate physical parameter estimation from videos under sparse-view settings. While many existing approaches rely on sequential or stage-wise optimization strategies (Li et al., 2023b; Zhong et al., 2024; Cai et al., 2024), as summarized in Table 1, such designs are prone to error accumulation across stages. In contrast, ProJo4D adopts a progressive joint optimization strategy, which significantly improves robustness and estimation accuracy in sparse-views.

## 3 ProJo4D

Our approach follows a multi-stage pipeline to estimate the constituting parameters: the appearance of a deformable object, initial physical states, and material properties. As is common in this domain, our pipeline begins with obtaining an initial 3D/4D representation. Our primary focus lies in the subsequent progressive joint optimization strategy designed to robustly solve inverse physics estimation from limited observations.

### 3.1 Motivation

**Physical Parameter Estimation.** Estimating the physical state and material parameters of an object from visual observations is a challenging inverse problem. This difficulty arises primarily because the whole system is non-linear and non-convex. Moreover, some material models, like non-Newtonian fluids, have non-differentiable but material-parameter-dependent branches. In addition, some of the physical parameters are strongly coupled, making it difficult to disambiguate their individual contributions from visual cues. All these difficulties necessitate careful design of optimization strategies to improve the chances of converging to a physically plausible and accurate solution.

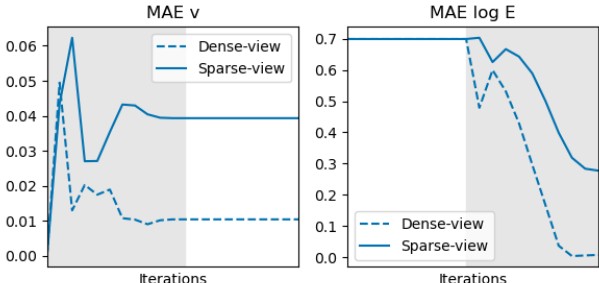

Figure 3: Comparison of error propagation in sequential optimization. The gray region marks the iterations during which the corresponding parameter is optimized: velocity (MAE-$v$, left) and material stiffness (MAE-$\log E$, right). Dense views reduce errors faster; sparse views accumulate more errors.

**Error accumulation in sparse vs. dense views under sequential optimization.** Most existing methods rely on sequential optimization (Li et al., 2023b; Cai et al., 2024; Zhang et al., 2024), where parameters are optimized in stages and estimates from earlier stages are fixed as inputs for later ones. While this strategy can help mitigate some challenges, it introduces a new problem: errors from earlier stages propagate and accumulate, with the effect being substantially worse for sparse-view settings.

For sparse views, the initial geometry estimation is considerably less accurate, and these errors cascade through subsequent optimization stages. This leads to large errors in estimating both initial states and material properties. Fig. 3 illustrates this phenomenon by plotting how mean absolute errors (MAE) in velocity ($v$) and material parameters ($\log E$) evolve during the sequential optimization. We exclude the shared initial stage, where the 3D/4D representation is constructed, and focus on the following two stages: velocity optimization (MAE $v$; left) and material parameter optimization (MAE $\log E$; right). With dense views, errors in the initial 3D/4D representation are smaller, leading to reduced error propagation in subsequent stages, whereas sparse views suffer from greater error accumulation across stages. For real-world deployments, capturing dense multi-view data with precisely calibrated and synchronized cameras is often impractical. Consequently, mitigating error accumulation and propagation becomes essential to extend the applicability of physics-based 4D reconstruction methods in real-world scenarios.

**Choice of optimization strategies: sequential vs. joint vs. progressive.** Fig. 4 illustrates how optimization strategies affect estimation accuracy and robustness across different object shapes and material models. We plot the error in material properties ($\log E$ for elastic (a) and $\theta_\alpha$ for sand (b)) and in future 4D state simulation (EMD) over optimization iterations. Sequential optimization, while common, suffers from significant error accumulation as shown in Fig. 4a, leading to high estimation errors.

An alternative is joint optimization, adopted by recent approaches such as Vid2Sim (Chen et al., 2025). This strategy can be effective for relatively simple models like elastic objects (Fig. 4a), but struggles with more complex systems such as sand (Fig. 4b) or non-Newtonian materials, where the optimization landscape is highly non-convex. In such cases, while initial geometry may already be close to ground truth, the physical parameters are typically far from accurate, causing optimization to stagnate in sub-optimal regions. A hybrid variant that performs joint optimization after sequential optimization attempts to address this but still inherits the limitations of the initial sequential stage.

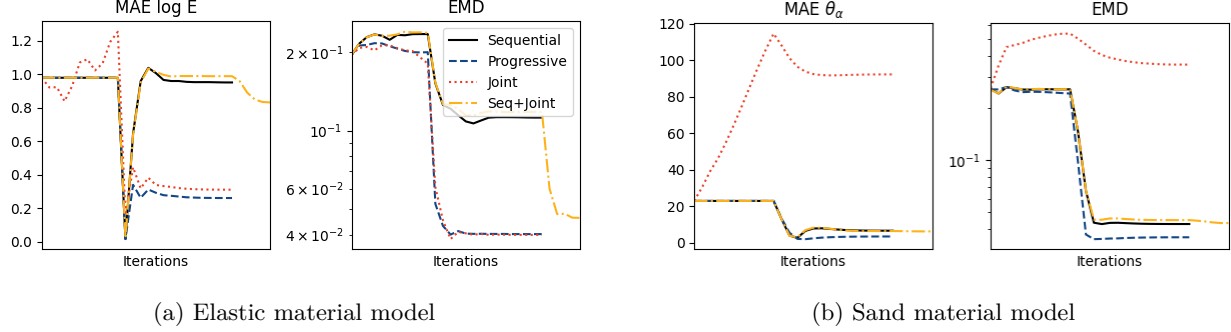

(a) Elastic material model  (b) Sand material model

Figure 4: Optimization trajectories for different strategies on elastic (a) and sand (b) materials. Progressive optimization (blue) achieves the best final accuracy while avoiding the instability of joint optimization (red), which diverges for complex materials like sand.

These findings highlight the need for more principled optimization strategies that generalize across diverse object shapes and material models. In the following section, we introduce our progressive joint optimization approach, which mitigates these issues, improving both performance and robustness.

## 3.2 Progressive Joint Optimization

To isolate the effect of our optimization strategy, we adopt the same 4D scene representation (3D Gaussian Splatting with deformation networks) and physics simulation framework (MPM) as GIC (Cai et al., 2024). Unlike recent approaches that introduce new neural architectures for material modeling (Zhao et al., 2025), our contribution lies solely in the progressive joint optimization strategy, which gradually expands the set of jointly optimized parameters across three stages (Fig. 2).

**Stage 0: Initial 3D / 4D Representation Learning.** Most existing pipelines, including ours, start with an initial 3D or 4D representation learning stage. Some methods (Li et al., 2023b; Zhong et al., 2024) learn a static 3D scene from the first image of each camera, while others (Cai et al., 2024) learn a full 4D representation from the multi-view image sequence. Our method is based on 4D representation learning to leverage 3D guidance during the following stages. The parameters for 4D representation are optimized by minimizing rendering losses over all frames and cameras:

$$\mathcal{L}_{img}(\hat{I}, I) = \lambda_{L1}\mathcal{L}_{L1}(\hat{I}, I) + \lambda_{SSIM}\mathcal{L}_{SSIM}(\hat{I}, I), \tag{4}$$

$$\hat{\mathbf{x}}_0^*, \hat{\Sigma}^*, \hat{c}^*, \hat{\sigma}^*, \psi^* = \operatorname*{arg\,min}_{\hat{\mathbf{x}}_0, \hat{\Sigma}, \hat{c}, \hat{\sigma}, \psi} \sum_{t \in \mathcal{T}} \sum_{c \in \mathcal{C}} \lambda_{img}\mathcal{L}_{img}\Big(\hat{I}_{t,c}, I_{t,c}\Big) + \lambda_\alpha \mathcal{L}_{L1}\Big(\hat{\alpha}_{t,c}, \alpha_{t,c}\Big), \tag{5}$$

where $\hat{I}_{t,c}$ and $\hat{\alpha}_{t,c}$ denote a rendered image and an alpha map, respectively (Sec. 2.1). $\mathcal{L}_{L1}$ and $\mathcal{L}_{SSIM}$ denote L1 loss and Structural similarity index measure (SSIM) loss, and $\lambda_{L1}, \lambda_{SSIM}$ are their corresponding loss weights.

**Stage 1: Initial Physical State Optimization.** This is the first stage of our progressive optimization strategy, focusing on estimating initial physical states $s$, such as the initial velocity $\hat{v}_0$. In this stage, we use the first few frames, following prior works (Li et al., 2023b; Zhong et al., 2024; Cai et al., 2024). This allows the optimization to focus on estimating initial velocity $\hat{v}_0$ before significant deformation or complex interactions take place, separating the influence of initial motion from material response. At this stage, Gaussian parameters are also optimized. As the initial velocity $v_0$ is the only physical state parameter in most existing benchmarks, we only optimize $\hat{v}_0$ by minimizing a combined loss over the first few frames $\mathcal{T}_k$:

$$\hat{v}_0^*, \hat{\mathbf{x}}_0^*, \hat{\Sigma}^*, \hat{c}^*, \hat{\sigma}^* = \operatorname*{arg\,min}_{\hat{v}_0, \hat{\mathbf{x}}_0, \hat{\Sigma}, \hat{c}, \hat{\sigma}} \lambda_{img} \sum_{t \in \mathcal{T}} \sum_{c \in \mathcal{C}} \mathcal{L}_{img}\Big(\tilde{I}_{t,c}, I_{t,c}\Big) + \lambda_{geo} \sum_{t \in \mathcal{T}} \mathcal{L}_{geo}\Big(\tilde{\mathbf{x}}_t, \hat{\mathbf{x}}_t\Big), \tag{6}$$

where $\mathcal{L}_{geo}$ is the bidirectional chamfer distance, which measures the distance to the closest point from both estimated and ground truth points. $\tilde{\mathbf{x}}_t$ and $\tilde{I}_{t,c}$ are positions from physics simulation $\mathcal{S}(\cdot)$ and their corre-

sponding rendered image (Sec. 2.1). $\hat{\mathbf{x}}_t$ denotes the extracted positions from the learned 4D representation using deformation network $\Delta(\cdot)$, as proposed in GIC (Cai et al., 2024).

**Stage 2: Full Joint Optimization.** After obtaining an improved estimate for physical states $s$, more specifically $v_0$, this stage progresses to include material parameters $\hat{\theta}_m$ during optimization. For this stage, we utilize data from all frames. We use the same optimization objective as the previous stage:

$$\hat{v}_0^*, \hat{\theta}_m^*, \hat{\mathbf{x}}_0^*, \hat{\Sigma}^*, \hat{c}^*, \hat{\sigma}^* = \underset{\hat{v}_0, \hat{\theta}_m, \hat{\mathbf{x}}_0, \hat{\Sigma}, \hat{c}, \hat{\sigma}}{\arg\min} \quad \lambda_{img} \sum_{t \in \mathcal{T}} \sum_{c \in \mathcal{C}} \mathcal{L}_{img}\Big(\tilde{I}_{t,c}, \, I_{t,c}\Big) + \lambda_{geo} \sum_{t \in \mathcal{T}} \mathcal{L}_{geo}\Big(\tilde{\mathbf{x}}_t, \hat{\mathbf{x}}_t\Big). \tag{7}$$

Table 2: Evaluation on Spring-Gaus (Zhong et al., 2024) dataset with sparse views (3 views). We measure 3D prediction accuracy of future states using Chamfer Distance (CD) and Earth Mover's Distance (EMD), image rendering quality of future states using PSNR and SSIM, and MAE of Young's modulus $E$ and Poisson's ratio $\nu$. For rendering quality evaluation, we used future images from all cameras.

| | method | apple | banana | chess | cream | cross | paste | torus | mean |
|---|---|---|---|---|---|---|---|---|---|
| CD ↓ | Spring-Gaus | 12.12 | 51.35 | 3.68 | 2.97 | 40.30 | 73.08 | 15.00 | 26.93 |
| | GIC | 2.13 | 8.37 | 7.51 | 8.16 | 2.51 | 81.24 | 2.81 | 16.11 |
| | MASIV | 1.02 | 2.23 | 5.15 | 2.42 | 4.75 | 2.33 | 1.51 | 2.77 |
| | GIC + ProJo4D | **0.19** | **0.12** | **1.37** | **1.54** | **0.38** | **6.93** | **0.65** | **1.60** |
| EMD ↓ | Spring-Gaus | 0.170 | 0.223 | 0.097 | 0.101 | 0.232 | 0.248 | 0.177 | 0.178 |
| | GIC | 0.090 | 0.106 | 0.139 | 0.135 | 0.084 | 0.263 | 0.081 | 0.128 |
| | MASIV | 0.053 | 0.051 | 0.124 | 0.088 | 0.102 | 0.110 | 0.046 | 0.082 |
| | GIC + ProJo4D | **0.054** | **0.024** | **0.066** | **0.052** | **0.031** | **0.142** | **0.031** | **0.057** |
| PSNR ↑ | Spring-Gaus | 17.03 | 15.79 | 13.85 | 14.62 | 11.24 | 10.94 | 13.01 | 13.78 |
| | GIC | 20.52 | 21.84 | 14.87 | 13.93 | 22.51 | 12.41 | 17.00 | 17.58 |
| | MASIV | 21.77 | 24.07 | 15.05 | 16.74 | 21.67 | 15.94 | 18.64 | 19.13 |
| | GIC + ProJo4D | **27.10** | **28.65** | **17.96** | **18.76** | **28.09** | **15.20** | **20.354** | **22.30** |
| SSIM ↑ | Spring-Gaus | 0.790 | 0.825 | 0.792 | 0.796 | 0.819 | 0.737 | 0.831 | 0.799 |
| | GIC | 0.868 | 0.910 | 0.826 | 0.795 | 0.889 | 0.772 | 0.892 | 0.850 |
| | MASIV | 0.884 | 0.930 | 0.826 | 0.849 | 0.890 | 0.852 | 0.922 | 0.879 |
| | GIC + ProJo4D | **0.930** | **0.959** | **0.886** | **0.885** | **0.943** | **0.852** | **0.933** | **0.913** |
| MAE $\log E$ ↓ | GIC | 0.1840 | 0.4639 | 0.1807 | 0.0838 | 0.3239 | **0.1380** | 0.2436 | 0.2311 |
| | GIC + ProJo4D | **0.0633** | **0.1519** | **0.0326** | **0.0336** | **0.0469** | 0.1705 | **0.2315** | **0.1043** |
| MAE $\nu$ ↓ | GIC | 0.1439 | **0.1049** | 0.0622 | 0.1407 | 0.0955 | 0.2209 | 0.4851 | 0.1790 |
| | GIC + ProJo4D | **0.0817** | 0.2237 | **0.0222** | **0.0295** | **0.0307** | **0.0928** | **0.1569** | **0.0911** |

## 4 Experiments

### 4.1 Experimental Settings

**Baselines.** We compare ours with the current state-of-the-art methods; PAC-NeRF (Li et al., 2023b), Spring-Gaus (Zhong et al., 2024), GIC (Cai et al., 2024), and Vid2Sim (Chen et al., 2025).

**Datasets.** We used the synthetic dataset from Spring-Gaus (Zhong et al., 2024), PAC-NeRF (Li et al., 2023b), and the GSO dataset from Vid2Sim (Chen et al., 2025). In the Spring-Gaus dataset, 7 distinct object shapes are provided, whereas the GSO dataset offers 12 shapes; both datasets only involve elastic materials with varying parameters. We used the PAC-NeRF synthetic dataset that comprises five different material models: elastic (Neo-Hookean), Newtonian fluid, non-Newtonian fluid, plasticine, and sand (Drucker-Prager), each sharing the same object shape but with different physical parameters. The dataset has a total of 45 scenes, and we report the mean and standard deviation for each material model. To evaluate how our proposed optimization strategy can improve performance in sparse-view settings, we selected only three cameras from the ten cameras available for both datasets: the second, sixth, and tenth cameras from each

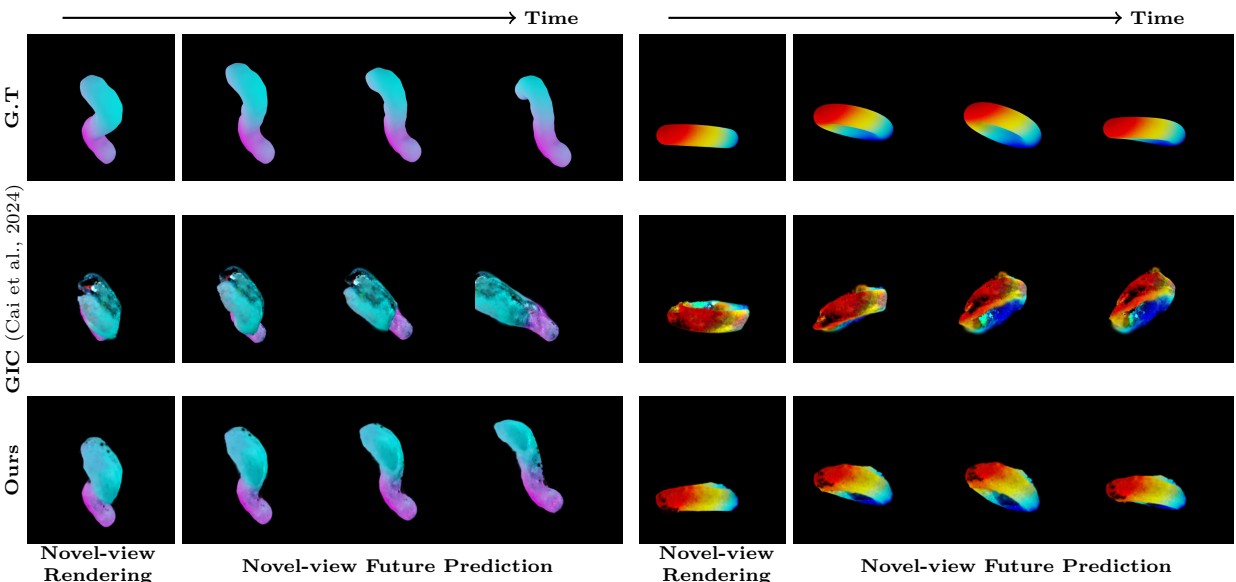

Figure 5: Visual comparison of ProJo4D (Ours) with GIC (Cai et al., 2024) for novel-view rendering and prediction in future timestep on the Spring-Gaus dataset, using sparse-view inputs. ProJo4D produces more consistent and physically plausible results across both current and future views.

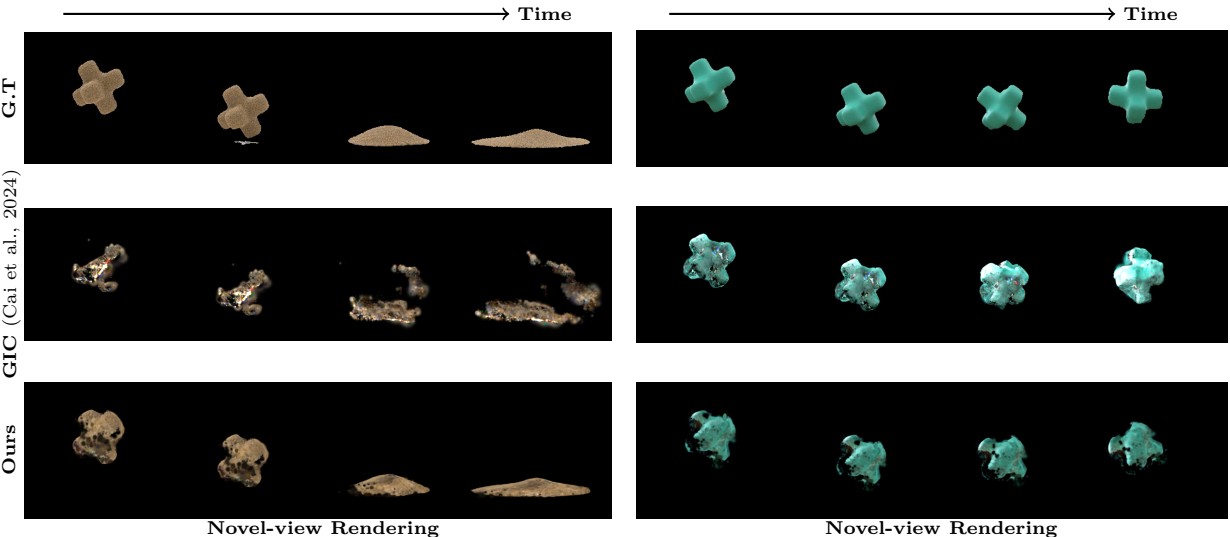

Figure 6: Visual comparison of ProJo4D (Ours) with GIC (Cai et al., 2024) for novel-view rendering on 'Sand' (left) and 'elastic' (right) materials from the PAC-NeRF dataset, using sparse-view inputs. ProJo4D reconstructs significantly better geometry than GIC.

scene using Spring-Gaus and PAC-NeRF datasets. For the GSO dataset, we used the same experimental settings as Vid2Sim. To compare with other baselines, we used the same train/test splits provided by each dataset.

For real-world evaluation, we use the Spring-Gaus dataset, which provides exactly three cameras for each scene. Following Spring-Gaus (Zhong et al., 2024), we optimize with the same data preprocessing steps. We train each model on the first 14 frames and test on the remaining 6 frames.

**Metrics.** To evaluate our method against state-of-the-art approaches, we adopt two categories of metrics: future state prediction and physical parameter estimation, following prior works (Li et al., 2023b; Zhong

Table 3: Evaluation on PAC-NeRF dataset. We report future prediction accuracy using Chamfer Distance (CD) and Earth Mover's Distance (EMD), and material estimation performance with mean absolute error.

| | | Elasticity | Newtonian | Non-Newtonian | Plasticine | Sand |
|---|---|---|---|---|---|---|
| CD ↓ | Spring-Gaus | 64.076 ± 107.271 | - | - | - | - |
| | GIC | 5.512 ± 3.311 | 0.537 ± 0.315 | 0.689 ± 0.398 | 2.012 ± 1.797 | 20.262 ± 43.360 |
| | ProJo4D | **0.913 ± 0.301** | **0.339 ± 0.108** | **0.473 ± 0.248** | **1.103 ± 0.948** | **0.264 ± 0.017** |
| | Full Joint | 1.318 ± 1.117 | 0.346 ± 0.095 | 8.104 ± 13.563 | 17.678 ± 18.170 | 53.564 ± 19.404 |
| EMD ↓ | Spring-Gaus | 0.267 ± 0.154 | - | - | - | - |
| | GIC | 0.126 ± 0.041 | 0.103 ± 0.007 | 0.040 ± 0.007 | 0.062 ± 0.027 | 0.122 ± 0.162 |
| | ProJo4D | **0.042 ± 0.007** | **0.039 ± 0.004** | **0.038 ± 0.005** | **0.053 ± 0.018** | **0.045 ± 0.006** |
| | Full Joint | 0.049 ± 0.023 | 0.040 ± 0.005 | 0.099 ± 0.073 | 0.124 ± 0.074 | 0.223 ± 0.020 |
| MAE $v_0$ ↓ | Spring-Gaus | 0.292 ± 0.057 | - | - | - | - |
| | GIC | 0.008 ± 0.004 | 0.009 ± 0.004 | 0.015 ± 0.008 | **0.010 ± 0.005** | 0.007 ± 0.004 |
| | ProJo4D | **0.007 ± 0.003** | **0.008 ± 0.002** | **0.005 ± 0.003** | 0.024 ± 0.056 | **0.005 ± 0.003** |
| | Full Joint | 0.020 ± 0.033 | 0.008 ± 0.004 | 0.080 ± 0.099 | 0.092 ± 0.102 | 0.046 ± 0.032 |
| MAE $\log(E)$ ↓ | Spring-Gaus | - | - | - | - | - |
| | GIC | 0.189 ± 0.217 | - | - | 1.597 ± 1.150 | - |
| | ProJo4D | **0.124 ± 0.099** | - | - | **0.742 ± 0.780** | - |
| | Full Joint | 0.216 ± 0.299 | - | - | 2.856 ± 2.196 | - |
| MAE $\nu$ ↓ | Spring-Gaus | - | - | - | - | - |
| | GIC | 0.123 ± 0.103 | - | - | 0.134 ± 0.112 | - |
| | ProJo4D | **0.048 ± 0.034** | - | - | **0.084 ± 0.029** | - |
| | Full Joint | 0.061 ± 0.053 | - | - | 0.075 ± 0.057 | - |
| MAE $\log(\mu)$ ↓ | Spring-Gaus | - | - | - | - | - |
| | GIC | - | **0.103 ± 0.125** | 0.869 ± 0.598 | - | - |
| | ProJo4D | - | 0.134 ± 0.175 | **0.491 ± 0.363** | - | - |
| | Full Joint | - | 0.294 ± 0.314 | 2.315 ± 1.100 | - | - |
| MAE $\log(\kappa)$ ↓ | Spring-Gaus | - | - | - | - | - |
| | GIC | - | 3.180 ± 1.085 | 0.725 ± 0.704 | - | - |
| | ProJo4D | - | **1.425 ± 1.148** | **0.462 ± 0.344** | - | - |
| | Full Joint | - | 3.312 ± 1.679 | 1.673 ± 1.918 | - | - |
| MAE $\log(\tau_Y)$ ↓ | Spring-Gaus | - | - | - | - | - |
| | GIC | - | - | **0.069 ± 0.069** | 0.327 ± 0.365 | - |
| | ProJo4D | - | - | 0.144 ± 0.071 | **0.144 ± 0.125** | - |
| | Full Joint | - | - | 1.839 ± 3.242 | 6.612 ± 7.739 | - |
| MAE $\log(\eta)$ ↓ | Spring-Gaus | - | - | - | - | - |
| | GIC | - | - | 0.519 ± 0.264 | - | - |
| | ProJo4D | - | - | **0.463 ± 0.244** | - | - |
| | Full Joint | - | - | 1.455 ± 2.263 | - | - |
| MAE $\theta_{fric}$ ↓ | Spring-Gaus | - | - | - | - | - |
| | GIC | - | - | - | - | 6.785 ± 8.458 |
| | ProJo4D | - | - | - | - | **4.998 ± 2.542** |
| | Full Joint | - | - | - | - | 67.893 ± 12.416 |

et al., 2024; Cai et al., 2024). For future state prediction, we measure the 3D discrepancy between simulated positions $\tilde{x}_t$ and ground-truth positions $x_t$ using Chamfer Distance (CD) and Earth Mover's Distance (EMD). We also assess the 2D rendering quality of predicted future states from both seen viewpoints (three in our experiments) and novel viewpoints using peak signal-to-noise ratio (PSNR) and structural similarity index measure (SSIM). For physical parameter estimation, we report mean absolute error (MAE), following previous literature (Li et al., 2023b; Zhong et al., 2024; Cai et al., 2024).

**Hyperparameters.** To focus on the optimization strategy, we use the same learning rates as GIC for both Spring-Gaus and the PAC-NeRF datasets. We also optimized with the same number of iterations for Stage 1 and 2 as GIC. We set the loss weights $\lambda_{img}$ and $\lambda_{geo}$ to $1/|\mathcal{C}|$ and 1.0, respectively (Eqs. 6 and 7).

Table 4: Future state prediction and material parameter estimation on the GSO dataset.

| | PSNR ↑ | MAE $\log E$ ↓ | MAE $\nu$ ↓ |
|---|---|---|---|
| PAC-NeRF | 20.11 | 2.50 | 0.21 |
| Spring-Gaus | 18.32 | - | - |
| GIC | 19.20 | 2.01 | 0.16 |
| Vid2Sim | 25.07 | 0.51 | **0.06** |
| GIC* | 21.90 | 0.59 | 0.07 |
| GIC* + ProJo4D | **26.80** | **0.31** | **0.06** |

Table 5: 2D future state prediction accuracies on Spring-Gaus real-world dataset (Zhong et al., 2024).

| | | bun | burger | dog | pig | potato | mean |
|---|---|---|---|---|---|---|---|
| PSNR ↑ | Spring-Gaus | 26.79 | 35.13 | 30.31 | 31.95 | 28.96 | 30.63 |
| | GIC | 32.14 | 36.89 | 33.35 | 32.30 | 35.02 | 34.05 |
| | GIC + ProJo4D | **37.35** | **39.01** | **36.07** | **38.90** | **40.18** | **38.30** |
| SSIM ↑ | Spring-Gaus | 0.986 | 0.995 | 0.993 | 0.994 | 0.989 | 0.991 |
| | GIC | 0.994 | 0.995 | 0.995 | 0.996 | 0.995 | 0.995 |
| | GIC + ProJo4D | **0.997** | **0.996** | **0.996** | **0.997** | **0.997** | **0.996** |

## 4.2 Results

**Synthetic Datasets.** We evaluate ProJo4D on three synthetic datasets covering diverse object shapes, appearances, and material models.

The Spring-Gaus dataset evaluates performance across diverse object shapes and appearances. As shown in Table 2, both Spring-Gaus and GIC degrade significantly under sparse views, especially on trajectory-sensitive scenes such as paste, whereas our method remains considerably more robust. Fig. 5 further illustrates that existing approaches fail to estimate geometry and physical parameters, critically failing to predict future trajectories. In contrast, our method remains robust with sparse views, achieving a tenfold reduction in Chamfer Distance, roughly half the Earth Mover's Distance on average, and a substantial PSNR boost ($17.58 \rightarrow 22.30$). MASIV achieves strong performance (CD: 2.77, EMD: 0.082) by introducing neural material models and additional loss function, but this requires around a day of training per elastic object compared to an hour for ProJo4D on an NVIDIA A6000. ProJo4D achieves better accuracy (CD: 1.60, EMD: 0.057) using the same scene representation and physics framework as GIC, with performance improvements attributable solely to our progressive joint optimization strategy. This demonstrates that careful optimization design can outperform architectural innovations while maintaining computational efficiency and physical interpretability.

The PAC-NeRF dataset evaluates the accuracy and robustness to different material models and their parameters. Table 3 and Fig. 6 show both quantitatively and qualitatively that our method outperforms GIC across different material models and different metrics and parameters. Both non-Newtonian and Plasticine have non-differentiable branches, which pose additional challenges to the progressive joint optimization strategy. Nevertheless, our method improves upon GIC in 4/5 parameters for Non-Newtonian and in 3/4 parameters for Plasticine, and shows strong improvement in 3D reconstruction (CD & EMD).

We also evaluate our method on the GSO dataset from Vid2Sim (Chen et al., 2025). To ensure a fair comparison, we reran GIC and ProJo4D using the same material model as Vid2Sim on the GSO dataset. Although Vid2Sim, PAC-NeRF, and GIC all adopt the Neo-Hookean material model for elastic objects, their exact formulations differ. The variant, GIC*, in Table 4 reports the performance of GIC with the identical material model as Vid2Sim. Results indicate that aligning the material models leads to improved performance for GIC. Our proposed method, ProJo4D, consistently demonstrates further performance improvements.

In summary, ProJo4D broadly improves over existing methods across different datasets, material models, and object shapes, with strong improvement in future trajectory prediction. In multi-parameter inverse problems, multiple parameter configurations can produce nearly indistinguishable dynamics in simulation, as physical parameters are often coupled. The future state prediction accuracy, which reflects the combined effect of all estimated parameters, is a more indicative measure of overall estimation quality. Examples are "banana" and "paste" in the Spring-Gaus dataset: while ProJo4D underperforms GIC in a single physics parameter, it achieves lower CD and EMD and higher PSNR in future prediction. ProJo4D not only improves parameter estimation on average but, more importantly, delivers better future state prediction accuracy, which is especially relevant for downstream applications such as simulation and digital twin construction.

**Real-world Dataset.** We evaluate ProJo4D on the real-world Spring-Gaus dataset to demonstrate practical applicability. Table 5 provides results on the Spring-Gaus real-world dataset. Since no ground truth three-dimensional mesh or material parameters are available, we evaluate using only two-dimensional metrics:

PSNR and SSIM. Because Spring-Gaus works exclusively with elastic objects, we use the elastic material model for both GIC (Cai et al., 2024) and our method. Consistent with the synthetic data results in Tables 2 and 3, our method outperforms other approaches in both PSNR and SSIM on real-world images. This demonstrates that our proposed optimization strategy significantly enhances estimation performance not only in synthetic but also in real-world settings, with additional visual results provided in the appendix.

Table 6: **Comparison with alternative optimization strategies.** Optimization order is denoted by X (position), A (appearance), S (velocity), and M (material parameters). "Sequential" is the GIC baseline. "Sequential+" increases iterations to match ProJo4D's budget. "Cyclic" strategies (N=4) repeat optimization stages with fewer iterations per cycle.

| | ProJo4D XA-XAS-XASM | Sequential XA-S-M-A | Sequential+ XA-S-M-A | Cyclic (XA-S-M)×4-A | XA-(S-M-A)×4 |
|---|---|---|---|---|---|
| CD ↓ | **1.60** | 16.11 | 16.72 | 14.63 | 3.20 |
| EMD ↓ | **0.057** | 0.128 | 0.135 | 0.136 | 0.085 |
| PSNR ↑ | **22.30** | 17.58 | 18.01 | 16.96 | 19.04 |
| SSIM ↑ | **0.913** | 0.850 | 0.854 | 0.834 | 0.882 |
| MAE $\log E$ ↓ | **0.1043** | 0.2311 | 0.1958 | 0.2547 | 0.1616 |
| MAE $\nu$ ↓ | **0.0911** | 0.1790 | 0.2984 | 0.2524 | 0.2686 |

### 4.3 Ablation Study

**Optimization Strategy Comparison.** As motivated in Sec. 3.1, the choice of optimization strategy significantly impacts performance. Table 3 shows that full joint optimization fails for complex materials like Non-Newtonian, Plasticine, and Sand, despite working reasonably for simpler materials. Here, we investigate whether alternative scheduling strategies can match ProJo4D's progressive approach.

Table 6 compares ProJo4D against sequential and cyclic strategies on the Spring-Gaus dataset. "Sequential" refers to the baseline GIC. To ensure a fair comparison, we also evaluate "Sequential+," which matches ProJo4D's total iteration count and the number of images used per parameter set (detailed configurations in the appendix, Table 9). We further evaluate cyclic optimization, where parameter sets are optimized one at a time over multiple cycles while maintaining a constant total iteration budget:

1. **(XA-S-M)×4-A**: Cycles through 4D learning, velocity, and material optimization 4 times, followed by appearance refinement.
2. **XA-(S-M-A)×4**: Performs 4D representation learning first, then cycles through velocity, material, and appearance optimization 4 times.

The results show that simply increasing the number of iterations (Sequential+) does not improve meaningfully, confirming that the improvement of ProJo4D stems from the optimization strategy. Cyclic optimization can outperform sequential strategies when applied after 4D representation learning (XA-(S-M-A)×4), but introducing it too early ((XA-S-M)×4-A) degrades performance, as the 4D representation is not yet sufficiently accurate. Nevertheless, cyclic optimization remains less stable and less accurate than ProJo4D's progressive joint optimization. Additional ablations on Stage 1 design choices are provided in the appendix.

**Robustness to Camera Views.** We additionally evaluate robustness with respect to the number of camera views in Table 7. As shown in the table, future prediction and material parameter estimation performance improve as the number of camera views increases. By adjusting only the parameter sets optimized at each stage, the performance remains robust even when the number of views decreases. We also report Init CD, the Chamfer Distance at the initial frame. Init CD shows that the error accumulation from geometry inherent in sequential optimization can be alleviated by our proposed optimization strategy.

**Generalization to Other Frameworks.** To demonstrate the applicability of ProJo4D to other methods, we evaluate it on the Spring-Gaus backbone, which differs from GIC in several design choices (e.g., anchors vs. Gaussians for simulation). We adapt the ProJo4D strategy by jointly optimizing velocity, positions, and appearance in the first stage, and all physical parameters in the second stage. With these modifications, Chamfer Distance for 4D future prediction improves from 24.54 to 11.89, compared to 12.83 with the same

Table 7: Ablation study on the impact of different numbers of camera views.

| | | Number of cameras | | | |
| --- | --- | --- | --- | --- | --- |
| | | 1 | 2 | 3 | 10 |
| Init CD ↓ | GIC | 27.08 | 0.60 | 0.40 | **0.10** |
| | + ProJo4D | **26.79** | **0.52** | **0.34** | 0.11 |
| Test CD ↓ | GIC | 107.20 | 12.00 | 16.11 | 0.95 |
| | + ProJo4D | **57.74** | **1.66** | **1.60** | **0.65** |
| PSNR ↑ | GIC | 11.26 | 16.57 | 17.58 | 22.98 |
| | + ProJo4D | **15.48** | **20.56** | **22.30** | **26.95** |
| MAE log $E$ ↓ | GIC | 0.5807 | 0.4951 | 0.2311 | 0.1286 |
| | + ProJo4D | **0.2527** | **0.1094** | **0.1043** | **0.0643** |
| MAE $\nu$ ↓ | GIC | 0.2194 | 0.2752 | 0.1790 | **0.0458** |
| | + ProJo4D | **0.1434** | **0.1374** | **0.0911** | 0.0654 |

structural modifications but without progressive optimization. The smaller gains compared to GIC are consistent with Spring-Gaus's constraints: fixed anchor connections and reduced 3D spatial flexibility limit the benefits of progressive joint optimization.

## 5 Conclusion

We introduced ProJo4D, a progressive joint optimization framework that incrementally expands the set of jointly optimized parameters. This strategy ensures robust estimation of geometry, appearance, and physical parameters under highly ambiguous, sparse-view inputs. Evaluations on benchmark datasets show that ProJo4D consistently outperforms state-of-the-art methods in 4D future state prediction, novel view rendering, and physical parameter estimation, demonstrating practical relevance.

While ProJo4D shows strong performance in 4D scene reconstruction and inverse physics estimation from sparse-view videos, it shares limitations common to existing methods. First, it cannot overcome fundamental challenges from underlying material models, such as non-differentiable, material-parameter-dependent branches, which require longer iterations and increase sensitivity to learning rates for some, including non-Newtonian fluids. Second, reliance on computationally intensive physics simulations results in high costs. Future work should explore accelerating simulations via neural surrogates or other lightweight methods. Our results highlight that optimization strategy is an important but often overlooked component in inverse physics pipelines, and we hope this work encourages further investigation in this direction.

### Acknowledgments

This work is supported by a National Institute of Health (NIH) project #R21EB035832 "Next-gen 3D Modeling of Endoscopy Videos".

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

## A    Optimization Trajectory Analysis

Fig. 7 provides an in-depth analysis of the optimization dynamics across different stages of ProJo4D. This figure visualizes how various metrics, including Earth Mover's Distance (EMD), Peak Signal-to-Noise Ratio (PSNR), and Mean Absolute Error (MAE) for material parameters, evolve through Stage 0 (initial 4D representation learning), Stage 1 (joint optimization of positions, appearance, and velocity), and Stage 2 (full joint optimization including material parameters).

The trajectory analysis demonstrates that by introducing physics-informed gradients during joint optimization, ProJo4D refines geometry more effectively than existing sequential methods. The joint optimization of positions alongside physical parameters in Stages 1 and 2 improves the quality of the initial point clouds, which in turn improves physical parameter estimation.

## B    Stage 1 Design Choices

In the main paper, we compare ProJo4D against alternative optimization strategies (sequential, cyclic). Here, we provide additional analysis on the design choices within Stage 1 of our progressive optimization pipeline. Specifically, we investigate which parameters to optimize in Stage 1 before transitioning to full joint optimization in Stage 2.

We conduct experiments on the PAC-NeRF dataset, which contains multiple material models. Because different material models involve distinct parameterizations, we report only metrics common across all models: Chamfer distance (CD) for future dynamics estimation and mean absolute error (MAE) for initial velocity. Table 8 compares different Stage 1 configurations: optimizing only velocity (S), only material parameters (M), both sequentially (SM), velocity jointly with positions and appearance (XAS, our default), material jointly with positions and appearance (XAM), and all parameters jointly (XASM, i.e., full joint optimization immediately after Stage 0).

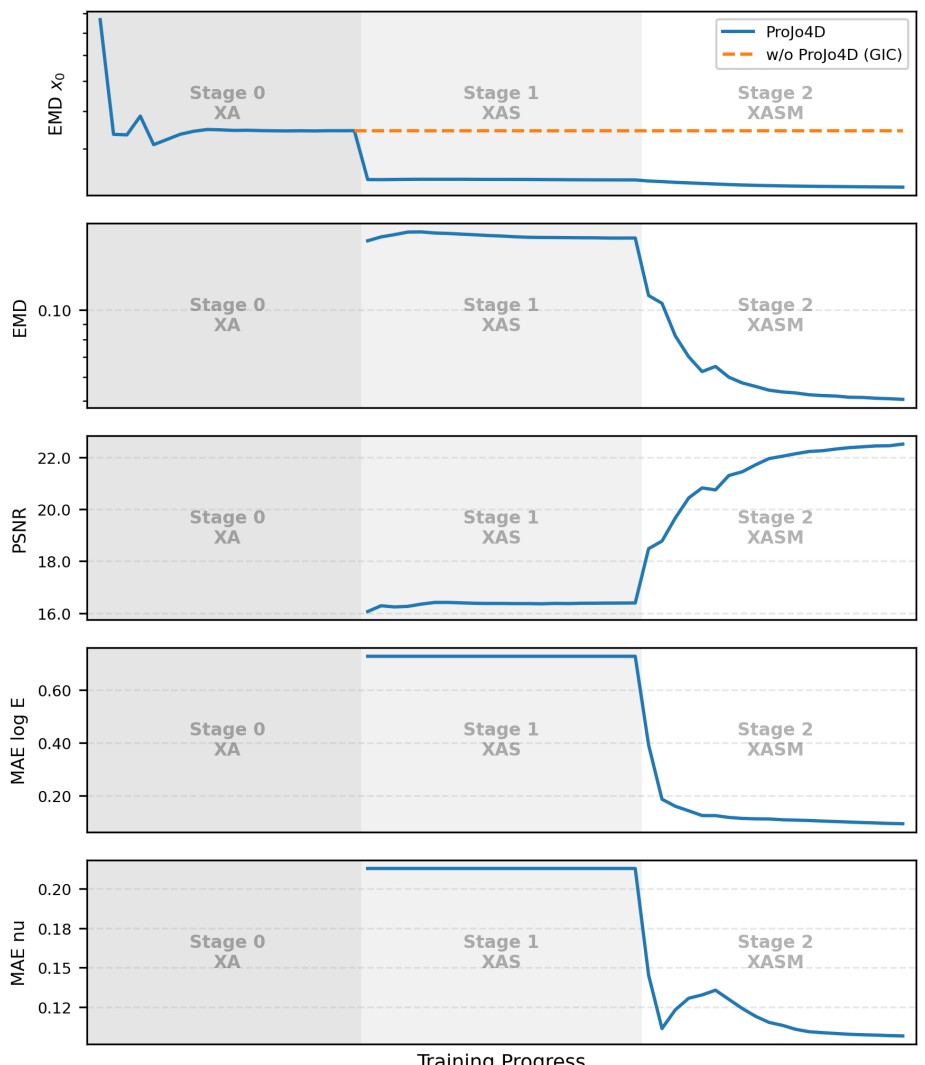

Figure 7: **Average optimization trajectories for Spring-Gaus scenes.** Background shading denotes Stage 0 (dark gray), Stage 1 (light gray), and Stage 2 (white). EMD and PSNR evaluate future-state prediction; MAE measures material parameter accuracy. $\text{EMD}_{x_0}$ indicates error in canonical space. Note: Stage 0 is rescaled for visualization due to its higher iteration count. By introducing physics-informed gradients during joint optimization, ProJo4D refines geometry more effectively than existing methods.

Our progressive method (XAS) demonstrates stability across different material models. Full joint optimization (XASM) achieves comparable performance to our method for relatively simple material models (Elastic and Newtonian), but shows significant degradation for more complex materials (Non-Newtonian, Plasticine, and Sand). This confirms that while full joint optimization can work for simpler models, it becomes unreliable as material complexity increases.

The results also show that optimizing velocity first (XAS) consistently outperforms material-first (XAM) across all materials. Initial velocity directly affects early-frame dynamics, making it easier to constrain from the first few frames, whereas material parameters govern longer-term deformation behavior that requires more frames to disambiguate.

Table 8: **Ablation on Stage 1 design choices across different material types.** We evaluate 3D future state prediction accuracy (CD) and initial velocity estimation (MAE $v_0$) on the PAC-NeRF dataset. Each row represents a different Stage 1 configuration, where X, A, S, and M denote positions, appearances, velocity, and material parameters, respectively. Bold values indicate best performance.

| | Stage 1 | Elastic | Newtonian | Non-Newtonian | Plasticine | Sand |
|---|---|---|---|---|---|---|
| CD ↓ | S | $0.953 \pm 0.295$ | $6.319 \pm 7.854$ | $9.668 \pm 4.543$ | $21.891 \pm 17.848$ | $2.727 \pm 0.531$ |
| | M | $1.020 \pm 0.314$ | $4.830 \pm 6.432$ | $9.205 \pm 3.559$ | $20.470 \pm 17.660$ | $4.067 \pm 2.469$ |
| | SM | $1.057 \pm 0.349$ | $4.896 \pm 4.254$ | $9.682 \pm 4.126$ | $20.434 \pm 17.976$ | $2.743 \pm 0.543$ |
| | **XAS (ProJo4D)** | $\mathbf{0.913 \pm 0.301}$ | $\mathbf{0.339 \pm 0.108}$ | $\mathbf{0.473 \pm 0.248}$ | $\mathbf{1.103 \pm 0.948}$ | $\mathbf{0.264 \pm 0.017}$ |
| | XAM | $1.053 \pm 0.339$ | $3.226 \pm 1.607$ | $9.164 \pm 4.587$ | $20.368 \pm 16.643$ | $2.457 \pm 0.359$ |
| | XASM (Full Joint) | $1.318 \pm 1.117$ | $0.346 \pm 0.095$ | $8.104 \pm 13.563$ | $17.678 \pm 18.170$ | $53.564 \pm 19.404$ |
| MAE $v_0$ ↓ | S | $\mathbf{0.007 \pm 0.003}$ | $0.074 \pm 0.045$ | $0.132 \pm 0.031$ | $0.102 \pm 0.091$ | $0.085 \pm 0.039$ |
| | M | $0.035 \pm 0.025$ | $0.089 \pm 0.045$ | $0.154 \pm 0.052$ | $0.131 \pm 0.107$ | $0.150 \pm 0.128$ |
| | SM | $0.013 \pm 0.009$ | $0.073 \pm 0.044$ | $0.149 \pm 0.086$ | $0.128 \pm 0.070$ | $0.095 \pm 0.034$ |
| | **XAS (ProJo4D)** | $\mathbf{0.007 \pm 0.003}$ | $\mathbf{0.008 \pm 0.002}$ | $\mathbf{0.005 \pm 0.003}$ | $\mathbf{0.024 \pm 0.056}$ | $\mathbf{0.005 \pm 0.003}$ |
| | XAM | $0.099 \pm 0.036$ | $0.125 \pm 0.045$ | $0.162 \pm 0.031$ | $0.173 \pm 0.091$ | $0.252 \pm 0.056$ |
| | XASM (Full Joint) | $0.020 \pm 0.033$ | $0.008 \pm 0.004$ | $0.080 \pm 0.099$ | $0.092 \pm 0.102$ | $0.046 \pm 0.032$ |

Table 9: **Training configurations for Table 6.** Batch sizes are in parentheses. For multi-view/multi-frame stages, we use: (number of cameras × number of frames).

| | ProJo4D | Sequential (GIC) | Sequential+ |
|---|---|---|---|
| Stage 0 | 40K (1) | 40K (1) | 47K (1) |
| Stage 1 | 100 (3×3) | 100 (3×3) | 200 (3×3) |
| Stage 2 | 100 (3×20) | 100 (3×20) | 100 (3×20) |
| Stage 3 | 0 | 40K (1) | 40K (1) |

## C  Hyperparameters

**Stage 0: 3D/4D Representation Learning.** To ensure a fair comparison, ProJo4D and the GIC baseline share identical 4D representations. Both are obtained after 40K iterations: a 3K-iteration warmup for static initialization followed by 37K iterations using deformation networks.

**Stages 1 & 2: Joint Optimization.** We adopt the same hyperparameters as GIC to isolate the performance gains from our progressive joint strategy.

**Spring-Gaus Dataset.** ProJo4D uses 100 iterations each for Stage 1 and Stage 2. While GIC requires a 30K-iteration "Stage 3" for appearance refinement, ProJo4D skips this because appearance is already optimized jointly during Stages 1 and 2.

**PAC-NeRF Dataset.** We maintain GIC's iteration schedule per material:

- **Elastic/Sand**: 100 for Stage 1, 150 for Stage 2

- **Newtonian**: 100 for Stage 1, 250 for Stage 2

- **Non-Newtonian**: 100 for Stage 1, 350 for Stage 2

- **Plasticine**: 100 for Stage 1, 300 for Stage 2

## D  Experimental Details for the GSO Dataset

While Vid2Sim, PAC-NeRF, and GIC all utilize the Neo-Hookean model for elastic objects, their stress formulations vary.

**Stress Formulation Variants.** PAC-NeRF and GIC define the Kirchhoff stress tensor $\tau$ as:

$$\tau_{PAC-NeRF} = \mu FF^T + (\lambda J - \mu)I, \tag{8}$$

where $F$ is the deformation gradient, $J = \det(F)$, and $\mu, \lambda$ are Lamé parameters. Conversely, Vid2Sim (following Simplicits) uses:

$$\tau_{Simplicits} = \mu FF^T + (\lambda(J - 1) - \mu)JI. \tag{9}$$

**Fair Comparison.** For a controlled comparison on the GSO dataset, we reran both GIC and ProJo4D using the Simplicits formulation (Eq. 9).

