# OpenReview forum: "ProJo4D: Progressive Joint Optimization for Sparse-View Inverse Physics Estimation"
_TMLR — Accepted by TMLR_

### Review · Reviewer_JQ52 · 2026-02-25

**Summary Of Contributions:**

The paper presents ProJo4D, a progressive joint optimization pipeline for inverse physics estimation from sparse multi-view videos. The method first learns a 4D scene representation, then jointly optimizes geometry, appearance, and initial velocity using early frames, and finally performs full joint optimization by introducing material parameters. By gradually expanding the set of optimized variables, ProJo4D enables physics-informed gradients to refine geometry while avoiding the instability of direct full joint optimization and the error accumulation of sequential pipelines. Without changing the underlying representation or physics simulator, this staged optimization alone leads to substantial improvements in future state prediction, rendering quality, and physical parameter estimation.

Key strengths include its simplicity, general applicability, and strong empirical results, while limitations include manually designed stage schedules,  and dependence on predefined material and physics models.

**Audience:**

Yes

**Audience Explanation:**

Researchers in neural rendering, differentiable physics, and 4D scene reconstruction would be interested, as the paper shows that a simple progressive optimization strategy can substantially improve sparse-view inverse physics estimation without changing model architectures.

**Claims And Evidence:**

Yes

**Claims Explanation:**

The paper shows that ProJo4D significantly improves sparse-view inverse physics estimation, reducing Chamfer Distance on Spring-Gaus (3 views) from 16.11 to 1.60 while increasing PSNR from 17.58 to 22.30 and improving material estimation accuracy (e.g., Young’s modulus MAE 0.23 → 0.10).

The authors also demonstrate that these gains come from the optimization strategy rather than architectural changes, as simply increasing iterations in the sequential baseline does not help, whereas progressive optimization does. Robustness over full joint optimization is evidenced on PAC-NeRF, where full joint collapses on sand (CD 53.56) but ProJo4D achieves CD 0.26. Finally, real-world results further confirm generalization, with PSNR improving from 34.05 to 38.30.

**Requested Changes:**

The proposed method relies on fixed, manually predefined iteration counts to switch between training stages. These hyperparameters are not sufficiently justified in the current submission. The authors are encouraged to provide additional justification, for example through ablation studies analyzing the sensitivity of performance to different stage lengths, or by discussing how these values generalize across datasets and material types.

The paper does not clearly describe the physics simulation model. While differentiable MPM is mentioned, its formal definition, assumptions, and limitations are not sufficiently explained. In particular, it is unclear how the method would handle objects composed of multiple materials, or whether spatially varying material properties are supported. Clarifying these aspects would help better define the scope and applicability of the proposed approach.

There are issues in the related work section where some citations appear as question marks, indicating formatting or reference errors. These should be corrected for clarity and completeness.

---

> ### Author Response · Authors · 2026-03-16
>
> We thank the reviewer for the constructive feedback and address each requested change below.
>
> **Sensitivity of Iteration Count Hyperparameters**
> Our iteration counts were set to match those of the baseline (GIC): the physics-optimization iterations in each stage equal those of the corresponding GIC stage, ensuring a fair comparison with no extra compute.
> To verify robustness to the number of iterations per stage, we ran an ablation on an elastic object (3-view):
>
> | Stage 1 (App. + Init. Vel.) | Stage 2 (Full Joint) | CD | PSNR | MAE log E |
> |:---:|:---:|:---:|:---:|:---:|
> | 50 | 100 | 0.629 | 20.57 | 0.203 |
> | 50 | 150 | 0.607 | 21.29 | 0.175 |
> | **100** | **100** | **0.624** | **20.56** | **0.191** |
> | 100 | 150 | 0.619 | 21.37 | 0.164 |
> | GIC | GIC | 1.52 | 18.28 | 0.222 |
>
> Our default setting (100+100, bolded) uses the same iterations per stage as GIC and only differs in which variables are optimized per stage. All configurations yield comparable metrics, indicating that performance is not sensitive to the particular stage schedule. The full per-material iteration schedules are provided in the appendix (Sec. A.3).
>
>
> **Physics Simulation Model (Differentiable MPM)**
> We expanded the differentiable MPM description in Section 2.2 of the revised manuscript. The updated text covers the scope of MPM and how its differentiable formulation enables gradient-based optimization via automatic differentiation. We also clarify the assumption of a single material with a known constitutive model per object, which is shared by existing methods and benchmarks in this domain (PAC-NeRF, Spring-Gaus, GIC, and MASIV).
>
> **Missing References**
> We thank the reviewer for pointing out the broken citations and have corrected them in the revised manuscript.
>
> We hope the revised manuscript, together with the iteration sensitivity ablation above, addresses the reviewer's concerns.

---

> > ### Author Response · Authors · 2026-03-23
> >
> > Thank you again for your thoughtful and detailed feedback. If you have any further questions or concerns regarding our rebuttal or the paper, we'd be happy to clarify or provide additional context.

---

### Review · Reviewer_SQ45 · 2026-02-28

**Summary Of Contributions:**

The paper introduces ProJo4D, a framework designed to estimate the 4D representation and physical parameters (such as initial velocity and material properties) of deformable objects using sparse multi-view videos.

ProJo4D propose a progressive joint optimization strategy. Instead of optimizing parameters one after another or all at once (which is often unstable due to non-convexity), ProJo4D gradually expands the set of jointly optimized parameters across three stages.

**Audience:**

Yes

**Audience Explanation:**

The paper is very interesting for the community. Unlike previous models that require dense views (10+ cameras), ProJo4D maintains high performance with as few as three cameras. Experiments cover a wide range of materials, including elastic objects, Newtonian and non-Newtonian fluids, plasticine, and sand.

**Broader Impact Concerns:**

No ethical concerns.

**Claims And Evidence:**

Yes

**Claims Explanation:**

Yes, but there are some aspects to be improved for the evaluation's completeness and overall quality. Please see the weaknesses.

**Requested Changes:**

Weaknesses

- The framework assumes the specific material model (e.g., elastic vs. plastic) is known a priori, which may limit its in-the-wild applicability where the material type is unknown. Given that the material model must be known beforehand, how does ProJo4D perform if the wrong model is selected (e.g., treating a non-Newtonian fluid as purely Newtonian)?

- The overall visual quality is not good, with more artifacts from the video. And the cases are almost the toy example, too simple; more complex cases and real captured complex cases should be evaluated to demonstrate the generalizability and validity.

- While it improves geometry, it still begins with an initial 3D/4D reconstruction stage (Stage 0). If this initial stage fails completely due to extreme sparsity or occlusion, the subsequent joint optimization might still struggle to recover. How does the quality of the initial Stage 0 reconstruction specifically impact the final convergence? Is there a minimum threshold of coarse accuracy required for the progressive joint optimization to take over successfully?

- While more efficient than some neural constitutive models like MASIV, the joint optimization over multiple stages still requires significant iterations and a differentiable physics simulator (MPM). The three-stage is more complex and very hard to follow and reproduce.

- The paper focuses on 3 views. It would be beneficial to see a breakdown analysis—at what point (e.g., 2 views or 1 view) does the progressive joint optimization finally fail compared to sequential methods?

- While the paper mentions computational efficiency relative to MASIV, a direct comparison of total training wall-clock time against the sequential GIC baseline would clarify the overhead of the joint stages.

- What about the failure cases and limitations discussion?

---

> ### Author Response · Authors · 2026-03-16
>
> We thank the reviewer for the positive feedback and detailed suggestions.
>
> **Material model assumption**
> This assumption is shared by prior works (PAC-NeRF, GIC, Vid2Sim), as stated in Sec. 2.2. To test the effect of model mismatch, we ran a subset of non-Newtonian PAC-NeRF scenes using a Newtonian material model. The table below reports Test CD on the same subset under both the correct and mismatched material models:
>
> | Material Model | GIC | Ours |
> |---|---|---|
> | Correct (non-Newtonian) | 0.5737 | 0.4092 |
> | Mismatched (Newtonian) | 0.8905 | 0.7385 |
>
> As expected, using the wrong constitutive law (material model) degrades performance. However, ProJo4D's progressive optimization still yields consistent improvement over GIC even under the mismatched model (0.7385 vs. 0.8905), suggesting the optimization strategy is beneficial regardless of the material model choice.
>
>
> **Visual quality**
> The datasets used in our evaluation (PAC-NeRF, Spring-Gaus, GSO) are standard benchmarks in the inverse physics literature and have also been adopted by recent work, such as MASIV (ICCV 2025). Among these, the GSO dataset (Tab. 4) contains more realistic object shapes from the Google Scanned Objects collection, and we further evaluate on real-world captured data in Tab. 5. We note that sparse-view reconstruction from only 3 views is inherently challenging, and some visual artifacts are expected in this setting. Nevertheless, ProJo4D substantially improves over baselines: PSNR increases from 17.58 (GIC) to 22.30 on a synthetic dataset (Tab. 2), and from 34.05 to 38.30 on real-world data (Tab. 5). We acknowledge that extending to more complex multi-object scenes is a valuable direction for future work, but out of scope for this work.
>
> **Complexity and reproducibility**
> Our method does not add extra stages. It changes which parameters are optimized at each existing stage. For reproducibility, we will release the code upon acceptance.
>
> **Training time**
> For an elastic object, compared to GIC, ProJo4D takes a similar total training time (both about an hour on an NVIDIA A6000).
> GIC can be slightly slower as it needs a separate, final stage for refining colors.
> For instance, on ‘torus’ from the SpringGaus dataset, ProJo4D takes around 60 minutes while GIC takes around 70 minutes to complete.
> Compared to MASIV, which requires around a day per object, ProJo4D and GIC are substantially faster.
>
> **Initial geometry and sparser views**
> We added Init CD and 1-view results to Tab. 7 in the revision. Init CD measures the Chamfer Distance of the initial geometry (at time $t=0$) after optimization, reflecting how well the method recovers the 3D shape. As the table shows, error accumulation is a persistent challenge regardless of the number of cameras. Our optimization strategy alleviates this issue, including monocular setting. Even at 1 view, ProJo4D reduces Test CD from 107.20 to 57.74 and improves PSNR from 11.26 to 15.48, showing that the progressive optimization is still beneficial on average.
>
> | | | Number of cameras | | | |
> |---|---|:---:|:---:|:---:|:---:|
> | | | 1 | 2 | 3 | 10 |
> | Init CD (lower=better) | GIC | 27.08 | 0.60 | 0.40 | **0.10** |
> | | + ProJo4D | **26.79** | **0.52** | **0.34** | 0.11 |
> | Test CD (lower=better) | GIC | 107.20 | 12.00 | 16.11 | 0.95 |
> | | + ProJo4D | **57.74** | **1.66** | **1.60** | **0.65** |
> | PSNR (higher=better) | GIC | 11.26 | 16.57 | 17.58 | 22.98 |
> | | + ProJo4D | **15.48** | **20.56** | **22.30** | **26.95** |
> | MAE log E (lower=better) | GIC | 0.5807 | 0.4951 | 0.2311 | 0.1286 |
> | | + ProJo4D | **0.2527** | **0.1094** | **0.1043** | **0.0643** |
> | MAE nu (lower=better) | GIC | 0.2194 | 0.2752 | 0.1790 | **0.0458** |
> | | + ProJo4D | **0.1434** | **0.1374** | **0.0911** | 0.0654 |
>
>
> **Failure cases and limitations**
> The main limitation is that our optimization strategy cannot compensate for severely degraded initial geometry. At 1 view, the initial geometry is severely degraded, and while our method slightly improves it (27.08 to 26.79), the reconstruction quality remains far from what denser views achieve. The sharp drop between 2 views and 1 view indicates that problems at this level, such as incomplete geometry with significant missing regions, cannot be resolved through optimization strategy alone and would require advances in the reconstruction stage itself.
> In practice, at 2+ views with a reasonable baseline between two cameras, the initial geometry provides a sufficient starting point for progressive optimization to succeed.

---

> > ### Author Response · Authors · 2026-03-23
> >
> > Thank you again for your thoughtful and detailed feedback. If you have any further questions or concerns regarding our rebuttal or the paper, we'd be happy to clarify or provide additional context.

---

### Review · Reviewer_9m7r · 2026-03-08

**Summary Of Contributions:**

This paper studies sparse-view inverse physics estimation for deformable objects. The authors argue that, under sparse views, standard sequential pipelines accumulate geometric errors early, whereas full joint optimization is unstable because the problem is non-convex and partially nondifferentiable. The proposed method, ProJo4D, keeps the GIC representation and simulator largely unchanged and introduces a new optimization schedule that progressively expands the set of jointly optimized variables across stages. The paper reports strong gains over GIC and Spring-Gaus on several benchmarks, and also includes ablations against full joint, cyclic, and stronger sequential baselines.

The main strength is that the paper isolates the optimization question and provides nontrivial ablations showing that the schedule is not arbitrary: progressive optimization beats sequential, sequential+, cyclic, and often full joint optimization on the authors’ benchmarks. The main weakness is that the novelty is very narrow. The paper itself states that the contribution lies solely in the optimization strategy, while reusing GIC’s 4D scene representation and physics framework. That makes the work feel more like a training recipe or optimization heuristic on top of an existing method than a substantively new system.

**Additional Comments:**

The paper’s best argument is its ablation suite. The paper convinced me that the schedule is not completely ad hoc. But for TMLR, I do not think “take GIC and change only the optimization order” is strong enough on its own, especially when the broader claims about efficiency and practicality are not fully supported by equally strong evidence.

**Audience:**

Yes

**Audience Explanation:**

The topic is relevant. Sparse-view inverse physics estimation sits at the intersection of differentiable simulation, neural scene representations, and system identification, and the question of optimization strategy is meaningful in this area.

**Claims And Evidence:**

No

**Claims Explanation:**

The paper provides reasonably convincing evidence for a narrow claim: on top of the GIC backbone, this progressive schedule helps under sparse-view settings. The ablations against full joint optimization, cyclic schedules, and a stronger sequential baseline support the claim that optimization order matters on these tasks. However, the contribution is explicitly only an optimization schedule layered on top of GIC, with no new representation, simulator, or learning formulation.

- Baselines are inconsistent across datasets, so the reader never gets apples-to-apples comparisons. Table 2 compares mainly against Spring-Gaus, GIC, and MASIV; Table 3 compares mainly against GIC and full joint; and Table 4 switches to a modified GIC* with Vid2Sim’s material formulation. This weakens the “state-of-the-art” narrative.

- The paper emphasizes computational efficiency and argues that MASIV has a higher memory/time cost, but the main paper does not provide a direct runtime or GPU-memory comparison table. The main result tables report only accuracy-style metrics.

- "consistent improvement” language might be a bit too strong. On PAC-NeRF materials, ProJo4D is not uniformly better on every physical parameter; for example, it is worse than GIC on Newtonian MAE log($\mu$) and worse than GIC on non-Newtonian MAE log($\tau_Y$).

**Requested Changes:**

Please review the accuracy of claims above and try to address some of the weaknesses, if possible.

---

> ### Author Response · Authors · 2026-03-16
>
> We thank the reviewer for the thoughtful evaluation and for acknowledging the value of our ablation study in demonstrating that the optimization schedule is principled.
>
> **Novelty and contribution**
> Our work deliberately isolates the optimization strategy to show its importance, a factor that prior work has overlooked in favor of architectural and representational changes.
> Prior works have advanced scene representations and physical modeling but have not systematically studied effective optimization despite the highly non-convex nature of the problem. While optimization alone does not address all challenges, the substantial gains (10x CD reduction on Spring-Gaus, 3-view) demonstrate that it is a critical and underexplored component, especially under sparse views.
> Our ablations (Tabs. 6, 7, Appendix Tab. 8) show that the proposed progressive approach consistently outperforms sequential, sequential+, cyclic, and full joint alternatives.
> The improvement also transfers to the Spring-Gaus backbone (Sec. 4.3), which uses a different physics simulation, confirming it is not specific to GIC.
>
> **Baselines**
> We acknowledge that our numerical evaluation tables on 3 different datasets, Spring-Gaus, PAC-NeRF, and GSO (Tab. 2-4), use different baselines. However, this is due to methodological constraints. Below, we describe why certain methods are absent in some tables:
> - Spring-Gaus method was designed for elastic objects only (spring-mass model), hence we only included it on the Spring-Gaussian and GSO datasets, in Tab. 2 & 4. During rebuttal, we further include results on only the elastic category in the PAC-NeRF dataset in Tab 3, where it still underperforms compared to ProJo4D (CD of 64.076 vs. 0.913). However, applying the spring-gaussian model to non-elastic objects in the PAC-NeRF dataset would be improper, and hence not included in Tab. 3.
> - Vid2Sim assumes zero initial velocity, which holds only for its own GSO dataset (Tab. 4). It cannot operate on Spring-Gaus and PAC-NeRF datasets since those have objects falling with non-zero initial velocity.
> - MASIV can be run on any objects; however, it takes a day per object (vs. ~1 hour for ours), making the full PAC-NeRF dataset (45 scenes) impractical. For the GSO dataset, MASIV does not provide official configurations or results for this benchmark. We attempted to run MASIV using its default configuration designed for elastic objects (from the Spring-Gaus dataset), but it frequently encountered Inf/NaN values during optimization, causing crashes before convergence. We attempted to stabilize training and searched over simulation-related hyperparameters, strictly without modifying the algorithm itself, but could only obtain valid results for 2 out of 12 objects, and our ability to perform long-range hyper-parameter search is also limited by the fact that each run requires ~1day of compute for each object. Due to the lack of official configurations and the numerical instability described above, we exclude MASIV from the main GSO comparison. Nevertheless, we report results for the two successful cases below, where our method outperforms MASIV by a clear margin.
>
> || Lion (PSNR) | Turtle (PSNR) |
> |-|-|-|
> | MASIV | 20.51 | 24.09 |
> | Ours | 25.17 | 28.50 |
>
> **Computational Costs**
> As requested, we added training time comparisons in the revision: ProJo4D takes ~1 hour per elastic object on an NVIDIA A6000, while MASIV takes around a day. For memory usage, both methods preallocate similar GPU memory, making exact comparisons difficult. To avoid confusion, we removed the memory claim and reported only wall-clock training time in the revision.
>
> **Accuracy of Improvement Claims**
> We revised "consistent improvement" to "broadly improves" in the manuscript. ProJo4D improves on the majority of physical parameters (3/3 for elastic, 2/3 for Newtonian, 4/5 for non-Newtonian, 3/4 for Plasticine, and 2/2 for sand) and shows substantial gains on trajectory prediction metrics (e.g., Newtonian CD: 0.537 to 0.339). We consider future-state prediction accuracy a more indicative overall measure, as different parameter configurations can produce similar dynamics due to coupling (Sec. 4.2).
>
> We hope that the revised manuscript, together with the additional baselines and runtime comparisons, addresses the reviewer's concerns.

---

> > ### Author Response · Authors · 2026-03-23
> >
> > Thank you again for your thoughtful and detailed feedback. If you have any further questions or concerns regarding our rebuttal or the paper, we'd be happy to clarify or provide additional context.

---

### Comment · Editors_In_Chief · 2026-05-31

On 5/30/2026, upon request of the authors, the EiCs replaced the camera ready version with a revision that corrects the format of the references.

---

### Decision · Action_Editor_ojgM · 2026-04-22

**Recommendation:** Accept as is

**Audience:**

Yes

**Audience Explanation:**

The paper will be of interest to the community given its focus on physically accurate neural rendering from sparse views, which is related to world models, an emerging area of research.

**Claims And Evidence:**

Yes

**Claims Explanation:**

The paper's main claim is that a progressive optimization strategy alone can enable stable estimation of physical parameters with resulting improvements in downstream rendering quality. The revised paper has provided additional results from baselines (or explained why they may not be applicable), as well as additional results using a mismatched material model, using sparser views and training time analysis. These additions addressed many of the concerns from the reviewers about the generality of the approach and fairness of comparisons. Ablation studies also provided good evidence for the effectiveness of the progressive optimization strategy. Reviewers lean positive about the paper after the rebuttal and revisions and I agree with them that the paper passes the TMLR criteria.